# Test-retest reliability of behavioral and computational measures of advice taking under volatility

Povilas Karvelis[1,2]*, Daniel J. Hauke[3], Michelle Wobmann[4], Christina Andreou[5], Amatya Mackintosh[4], Renate de Bock[4], Stefan Borgwardt[5], Andreea O. Diaconescu[1,2,6,7]

1 Krembil Centre for Neuroinformatics, Centre for Addiction and Mental Health (CAMH), Toronto, ON, Canada, 2 Department of Psychiatry, University of Toronto, Toronto, ON, Canada, 3 Centre for Medical Image Computing, Department of Computer Science, University College London, London, United Kingdom, 4 Department of Psychiatry (UPK), University of Basel, Basel, Switzerland, 5 Department of Psychiatry and Psychotherapy, Translational Psychiatry, University of Lubeck, Lubeck, Germany, 6 Institute of Medical Sciences, University of Toronto, Toronto, ON, Canada, 7 Department of Psychology, University of Toronto, Toronto, ON, Canada

* povilas.karvelis@camh.ca

**Data Availability Statement:** The data is available at https://osf.io/a6v8w. The code is available at https://github.com/povilaskarvelis/compi_ioio_reliability.

## Abstract

The development of computational models for studying mental disorders is on the rise. However, their psychometric properties remain understudied, posing a risk of undermining their use in empirical research and clinical translation. Here we investigated test-retest reliability (with a 2-week interval) of a computational assay probing advice-taking under volatility with a Hierarchical Gaussian Filter (HGF) model. In a sample of 39 healthy participants, we found the computational measures to have largely poor reliability (intra-class correlation coefficient or $ICC < 0.5$), on par with the behavioral measures of task performance. Further analysis revealed that reliability was substantially impacted by intrinsic measurement noise (indicated by parameter recovery analysis) and to a smaller extent by practice effects. However, a large portion of within-subject variance remained unexplained and may be attributable to state-like fluctuations. Despite the poor test-retest reliability, we found the assay to have face validity at the group level. Overall, our work highlights that the different sources of variance affecting test-retest reliability need to be studied in greater detail. A better understanding of these sources would facilitate the design of more psychometrically sound assays, which would improve the quality of future research and increase the probability of clinical translation.

## Introduction

Computational modelling is a powerful tool for studying complex systems such as the brain. Combining computational models with cognitive tasks allows one to measure various latent neurocomputational variables (e.g., learning rate, reward sensitivity, sensory precision, decision noise) that would otherwise be unobservable. There is plenty of enthusiasm that such computational measures may become a new basis for psychiatric nosology and treatment personalization [1–7].

**Funding:** PK is supported by Canadian Institutes of Health Research (CIHR) Fellowship (472369). AOD is supported by Swiss National Science Foundation and the Krembil Foundation (1000824). The funders played no role in the study design, data collection and analysis, decision to publish, or preparation of the manuscript.

**Competing interests:** The authors have no competing interests to declare.

However, for the computational measures to be sufficiently informative at the individual level—whether for studying individual differences in the lab (e.g., correlational analyses with symptoms) or for personalized care in the clinic (e.g., treatment response prediction)—they must have excellent psychometric properties (reliability and construct validity). Up until recently, the psychometric properties of computational measures have received little attention [8]. Emerging empirical evidence indicates that both reliability and construct validity are often poor [9], but the psychometric properties of many currently used computational assays remain unknown and require further investigation.

Here we investigate test-retest reliability of a computational assay probing advice-taking under volatility. The assay is comprised of a social advice-taking task and a Hierarchical Gaussian Filter (HGF) model [10]. The task has been designed to probe how people infer the intentions of others (helpful vs. unhelpful) in the context where the intentions themselves are changing over time. This allows studying the dynamics of how people learn to trust or distrust others. Combining the task with the HGF model further helps to infer various cognitive variables of interest, in particular as it relates to participants' estimation of how volatile the adviser is and whether they are likely to be helpful or unhelpful at any given moment. Variants of this assay have been used to study multiple disorders that exhibit difficulties in correctly inferring the intentions of others, including schizophrenia [11, 12], autism [13], and borderline personality disorder [12], as well as aging [14], but the reliability of the measures afforded by the assay has not yet been studied.

A more general aim of this paper is to examine the different sources of variance that underlie poor test-retest reliability in more detail—something that is rarely done when test-retest reliability of computational measures is assessed [9]. Understanding what drives low test-retest reliability is the first step towards designing more psychometrically sound assays.

## Methods

### Participants

A total of 43 participants were recruited from the general population via online advertisements and advertisements in public places in Basel, Switzerland. Four participants completed only a single task session, leaving 39 participants for the analysis presented in this paper. The remaining sample included 20 female and 19 male participants, with an average age of 28.85 (SD = 4.03). All participants provided informed written consent. The study was approved by the local ethics committee (Ethikkommission Nordwest- und Zentralschweiz, no. 2017–01149) and conducted in accordance with the latest version of the Declaration of Helsinki. All data collection was done between November 1st 2017 and July 31st 2022.

### Study design

The data analyzed here was collected as part of a larger study investigating first-episode psychosis patients and individuals at clinical high-risk for psychosis [11]. Each participant completed the cognitive task twice, 2 weeks apart. One of the sessions was combined with fMRI and one with EEG measurements (the order of fMRI and EEG was randomized across participants). In this paper, however, we will focus solely on the behavioral data and only on the 39 healthy controls.

### The task

Participants engaged in a probabilistic learning task (Fig 1) where they had to infer on the changing intentions of an adviser and decide how much to rely on their advice; for previous

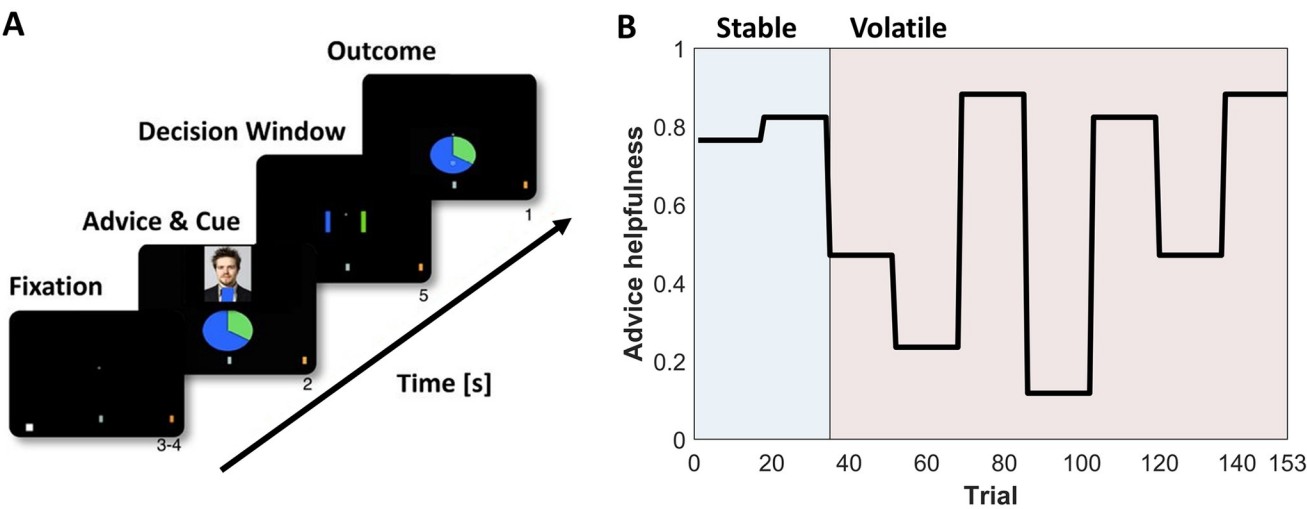

**Fig 1. The task.** (**A**) The trial structure: On each trial, participants are presented with two pieces of information: a social cue (a recording of an adviser holding up one of the colors), a non-social cue (a pie chart) indicating true winning probabilities of the two options. (**B**) Probability of adviser's fidelity (helpfulness) across the task.

studies using this task see [10, 11, 15]. The task had two phases: a stable phase with consistent helpful advice and a volatile phase where the adviser's intentions changed rapidly (Fig 1B). On each trial, participants predicted the outcome of a binary lottery using information from a non-social cue (displaying true winning probabilities) and a prerecorded video recommendation from a human adviser who had previously attempted to help or deceive a player in a human-human interaction. The participants were informed that the adviser had privileged information about the outcome, but not complete information, and that inaccurate advice could be due to mistakes of the adviser or the adviser having a different incentive structure that could motivate them to provide misleading advice during the experiment. In total, there were 153 trials.

In addition to making choices on every trial, participants were also prompted to explicitly report how helpful the adviser was at multiple times throughout the task. They were asked "How do you feel about the advice you are receiving right now?" on trials 14, 49, 73, 99, and 134, and responded by choosing among "helpful", "random", and "misleading". To probe their prior expectations, before starting the task (on trial 1), participants were also asked "How do you think the advice you are about to receive will be?" and had to respond by choosing among the same three options.

## Behavioral measures

For the behavioral analysis we computed standard behavioral measures. This included: 1) Total accuracy, which was defined as the fraction of trials in which participants chose the option with the higher reward probability. This measure captures the overall level of performance. 2) Advice taking probability, which was defined as the fraction of trials in which participants acted in accordance with the social cue. This measure captures the tendency to rely on advice, and we computed it separately for stable and volatile phases of the task, as well as for the whole task. 3) Win-stay probability, which was defined as the fraction of trials in which participants kept the same strategy (trusting the advice or not) following a winning trial. 4) Lose-switch probability, which was defined as the fraction of trials in which participants

switched their strategy (trusting the advice or not) following a losing trial. Both win-stay and lose-switch measures capture how quickly participants learn from positive and negative feedback, respectively. However, it is a rather rough proxy for the learning rate, because we expect it to dynamically change depending on how volatile the adviser's strategy is perceived to be at any given point in the task—to capture these dynamics more explicitly we next turn to computational modelling.

## Computational modelling

**Hierarchical Gaussian Filter.** To model participants' behavior in the task we used the same analysis pipeline as in the previous work [11], using a 3-level Hierarchical Gaussian Filter (HGF) [16, 17]. This modelling framework assumes that participants infer a hierarchy of hidden states ($x_1$, $x_2$, and $x_3$) that characterize outcome probabilities and their temporal dynamics. The participants' beliefs about the hidden states at each level $i$ on trial $k$ are denoted as $\mu_i^{(k)}$. In this task, the states that participants need to infer based on the inputs are structured as follows: the lowest level state represents the accuracy of the advice, which can be either accurate ($x_1^{(k)} = 1$) or inaccurate ($x_1^{(k)} = 0$). The relationship between this state and the second level state ($x_2^{(k)}$), representing the adviser's fidelity or tendency towards helpful advice, is described by a Bernoulli distribution linked to the level above with a unit sigmoid transformation:

$$p(x_1^{(k)}|x_2^{(k)}) = s(x_2^{(k)})^{x_1^{(k)}}(1 - s(x_2^{(k)}))^{1-x_1^{(k)}} \sim \text{Bernoulli}(x_1^{(k)}; s(x_2^{(k)})), \tag{1}$$

with $s$ being a sigmoid function:

$$s(z) = \frac{1}{1 + e^{-z}}. \tag{2}$$

The second level state ($x_2^{(k)}$) is modeled as a normal distribution with a mean ($x_2^{(k-1)}$) and variance determined by the volatility of the adviser's intentions ($x_3^{(k)}$):

$$p(x_2^{(k)}|x_2^{(k-1)}, x_3^{(k)}, \kappa_2, \omega_2) \sim \mathcal{N}(x_2^{(k)}; x_2^{(k-1)}, \exp(\kappa_2 x_3^{(k)} + \omega_2)) \tag{3}$$

where $\omega_2$ is the evolution rate that shapes tonic learning, and $\kappa_2$ is the coupling strength between the second and third levels shaping phasic learning. The third level state ($x_3^{(k)}$) represents the volatility of the adviser's intentions and is also modeled as a normal distribution:

$$p(x_3^{(k)}|x_3^{(k-1)}, \vartheta) \sim \mathcal{N}(x_3^{(k)}; x_3^{(k-1)}, \vartheta) \tag{4}$$

where $\vartheta$ is meta-volatility at the third level.

Following previous work [11, 18], this part of the model was augmented with an additional tendency to revert back to equilibrium beliefs about environmental volatility, akin to an Ornstein-Uhlenbeck process in discrete time [19]:

$$p(x_3^{(k)}|x_3^{(k-1)}, \vartheta, \phi_3, m_3) \sim \mathcal{N}(x_3^{(k)}; x_3^{(k-1)} + \phi_3(m_3 - x_3^{(k-1)}), \vartheta), \tag{5}$$

where $\phi_3$ is the drift rate and $m_3$ is the equilibrium point towards which the state moves over time.

After each observation, the model updates inferences about each of the states. The HGF update equations can be derived using a variational approximation, where the belief update at

each level depends on the precision from the level below and the prediction error:

$$\Delta\mu_i^{(k)} \propto \frac{\hat{\pi}_{i-1}^{(k)}}{\pi_i^{(k)}}\delta_{i-1}^{(k)}, \tag{6}$$

where $\mu_i^{(k)}$ is the expectation or the belief at trial $k$ and level $i$ of the hierarchy, $\hat{\pi}_{i-1}^{(k)}$ is the precision (inverse of the variance) from the level below, $\pi_i^{(k)}$ is the updated precision at the current level, and $\delta_{i-1}^{(k)}$ is the prediction error from the level below (the difference between the expected and the observed outcome).

Inferred states about the world are translated into decisions (i.e., to go with or against the advice) via the response model:

$$b^{(k)} = \zeta\hat{\mu}_1^{(k)} + (1-\zeta)c^{(k)}, \tag{7}$$

where $\zeta$ captures the relative weighing of the advice compared to the non-social cue, $c^{(k)}$ is the outcome probability indicated by the pie chart, transformed to reflect the colour probability indicated by the adviser, while $\hat{\mu}_1^{(k)}$ is the belief that the adviser is providing accurate advice.

The probability that a participant follows the advice ($y = 1$) can then be described by a sigmoid transformation of the integrated belief $b$:

$$p(y = 1|b) = \frac{b^\beta}{b^\beta + (1-b)^\beta}, \tag{8}$$

with

$$\beta = \exp(-\hat{\mu}_3^{(k)} + v). \tag{9}$$

where $\hat{\mu}_3^{(k)}$ is the current estimate of the volatility of the advisers' intentions, and $v$ is decision noise that is independent of the perception of volatility and is constant across trials.

In summary, the HGF models participants' behavior in the social learning task by inferring hidden states and updating them based on the outcome of each trial. The mean-reverting HGF additionally incorporates a drift in volatility estimates towards a subject-specific equilibrium (Fig 2). The response model integrates beliefs and cues to make decisions, with the noise level determined by the perception of volatility.

**Model space.**   In total, we have tested 2 variants of the HGF: standard HGF, (Eq 4) (model 1), and HGF with the mean-reverting process at the third level, (Eq 5) (model 2). For each of these variants, we included an additional control model (model 1b and model 2b), in which all parameters of the perceptual model were fixed to parameter values of an ideal Bayesian observer optimised based on the inputs alone.

The models were implemented in Matlab R2020a using HGF 3.0 open-source toolbox included in Translational Algorithms for Psychiatry-Advancing Science (TAPAS) [20] software collection https://github.com/translationalneuromodeling/tapas/releases/tag/v3.0.0). The code implementing the models can be found in 'tapas_hgf_binary.m' function for model 1 and 'tapas_hgf_ar1_binary.m' function for model 2.

**Model fitting.**   Following the modelling pipeline in Hauke et al. [11], meta-volatility $\vartheta$ was fixed to 0.5, and the drift rate parameter $\phi_3$ was fixed to 0.1, which was done to reduce the number of free parameters to help with model convergence. The remaining parameters were free, including: the evolution rate at the second level $\omega_2$, the coupling strength between the second and third level $\kappa_2$, prior expectations before seeing any input about the adviser's fidelity $\mu_2^{(0)}$ and the volatility of the adviser's intentions $\mu_3^{(0)}$, the relative weighing of the advice compared to the non-social cue $\zeta$, decision noise $v$, and volatility equilibrium $m_3$.

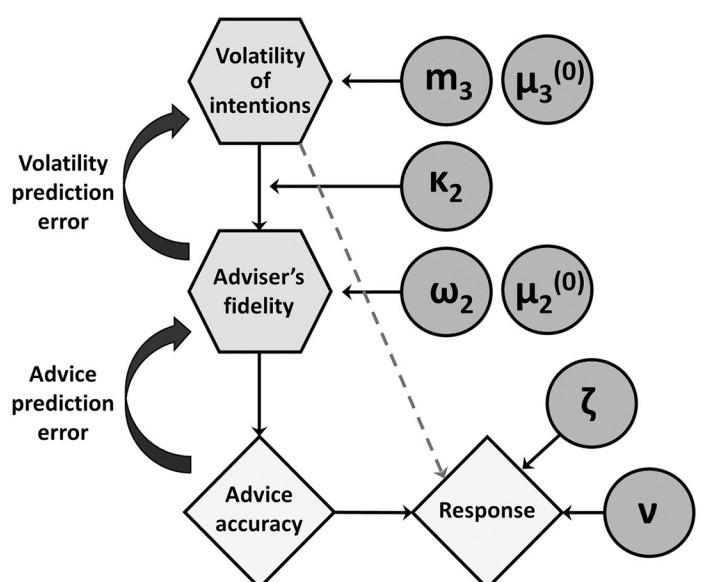

**Fig 2. Graphical representation of the mean-reverting 3-level HGF.** The highlighted parameters are free parameters. $m_3$ is volatility equilibrium point towards which the agent slowly drifts irrespective of empirical observations. $\mu_3$ and $\mu_2$ are the priors about advice volatility and fidelity, respectively, that the agent has before starting the task. $\kappa_2$ is the phasic learning rate which regulates how much the inferences about the volatility (at the third level) modulate learning about adviser's fidelity (at the second level). $\omega_2$ is the baseline leaning rate (evolution rate), which regulates the learning rate at the second level independent from the estimates at the third level. $\zeta$ parameter captures how much participants rely on the social cue relative to the non-social cue. Finally, $v$ captures intrinsic noise at the stage of decision-making that is unrelated to inferred volatility of the adviser.

Model fitting (i.e., finding parameter values with the highest posterior probability) was done using quasi-Newton optimization algorithm. The priors for all model parameters were based on previous work, the details of which can be found in Table 1 in Hauke et al. [11]. We briefly summarize them here in Table 1.

**Model selection.** To arbitrate among the models, we performed random-effects Bayesian model selection [21, 22], using VBA toolbox [23] (https://mbb-team.github.io/VBA-toolbox/). We present protected exceedance probabilities (PEP), which quantify the likelihood of a model being more plausible than any other model in the model space, while accounting for the possibility that model differences could arise by chance alone [21, 22]. Additionally, we report relative model frequencies or *f* as an effect size measure, representing the probability that a randomly selected participant would be best accounted for by a specific model.

**Parameter and model recovery.** To better understand test-retest reliability results, we also performed parameter recovery. Parameter recovery involves generating synthetic data of task performance using a range of model parameter values and then fitting the same model to that data [24, 25]. This procedure allows us to assess how uniquely different sets of parameter values map onto specific behavioral patterns, and vice versa. The similarity between the 'true' and the 'recovered' parameters indicates how reliable the estimates would be under ideal

**Table 1. Priors on free model parameters.** Prior means and their respective variances are denoted in brackets, and for some parameters followed by an upper bound: (mean, variance), upper bound. See Fig 2 for a detailed explanation of each parameter.

| | $\kappa_2$ | $\omega_2$ | $\mu_2^{(0)}$ | $\mu_3^{(0)}$ | $\zeta$ | $v$ | $m_3$ |
|---|---|---|---|---|---|---|---|
| **model 1** | (0, 1), 1 | (−2, 4) | (0, 1) | (1, 1) | (0, 1), 1 | (log(48), 1) | |
| **model 2** | (0, 1), 1 | (−2, 4) | (0, 1) | (1, 1) | (0, 1), 1 | (log(48), 1) | (1, 1) |

conditions: when task performance stays exactly the same across instances. As such, it is constrained only by the intrinsic measurement error and can be considered to provide an upper bound on test-retest reliability, because the latter would be further affected by within-subject changes in task performance [9].

For simulating task data, we simply used the same parameter estimates obtained from empirical data. This means that we had 39 synthetic agents and each of them completed 153 task trials. Due to the stochasticity in decision making, we reran this procedure 20 times with different random seeds. We report parameter recovery results for one of the runs as well as averaged cross the 20 runs.

To make sure that our models are recoverable, we repeated this procedure for all other models, while also fitting each model to the data generated by the other models. We then performed random-effects Bayesian model selection (based on the free energy approximation of the Bayesian model evidence) on each of the datasets and averaged the resulting protected exceedance probabilities across the 20 runs to produce a confusion matrix to assess model recovery.

### Statistical analysis

Test-retest reliability of the behavioral and computational measures was assessed using intra-class correlation coefficients (ICCs). More specifically, we computed ICCs(A,1): two-way mixed, single-measure, absolute agreement ICC [26, 27]. Note that in the [28] notation, this would be denoted as ICC(2,1). Calculations were performed using *icc* function from *irr* package (version 0.84.1) in R (version 4.2.2). For referring to different ranges of ICC values, we chose [29] labelling guidelines: poor $< 0.5$, moderate $0.5$–$0.75$, good $0.75$–$0.9$, and excellent $> 0.9$. Although note that these labels are somewhat arbitrary [9, 30].

Alongside ICCs, we also report Pearson's correlation coefficients in order to highlight when and how much these metrics diverge. Pearson's *r* is not sensitive to offset and scaling errors, which makes it a less suitable metric for assessing reliability [9]. However, as we will see, in practice this might be rarely a problem. Furthermore, we will use the comparison between Pearson's *r* and ICC to gauge the impact of practice effects on test-retest reliability.

Practice effects were investigated using paired t-tests, comparing the obtained measures between the two time points. Note that this addresses only fixed effects (e.g., people getting better at the task or less engaged with the task on average), but does not account for potential idiosyncratic practice effects. Given the limitations of frequentist statistics, wherever appropriate we computed Bayesian Factors [31] to test the strength of evidence for the null hypothesis ($BF_{01}$).

To make our findings more robust, we performed the above tests by excluding extreme outliers, which were based on the conventional Tukey's definition: as being below $Q1 - 3 * IQR$ or above $Q3 + 3*IQR$.

### Results

### Test-retest reliability of behavioral measures

First, we analyzed test-retest reliability of standard behavioral measures (Fig 3). We found most measures to be in the poor reliability range ($ICC < 0.5$), except for accuracy ($ICC = 0.61$) and win-stay ($ICC = 0.52$). Comparing both absolute ICC and Pearson's correlation for each of the measures also revealed that both indices provide similar results (with the largest difference being 0.03).

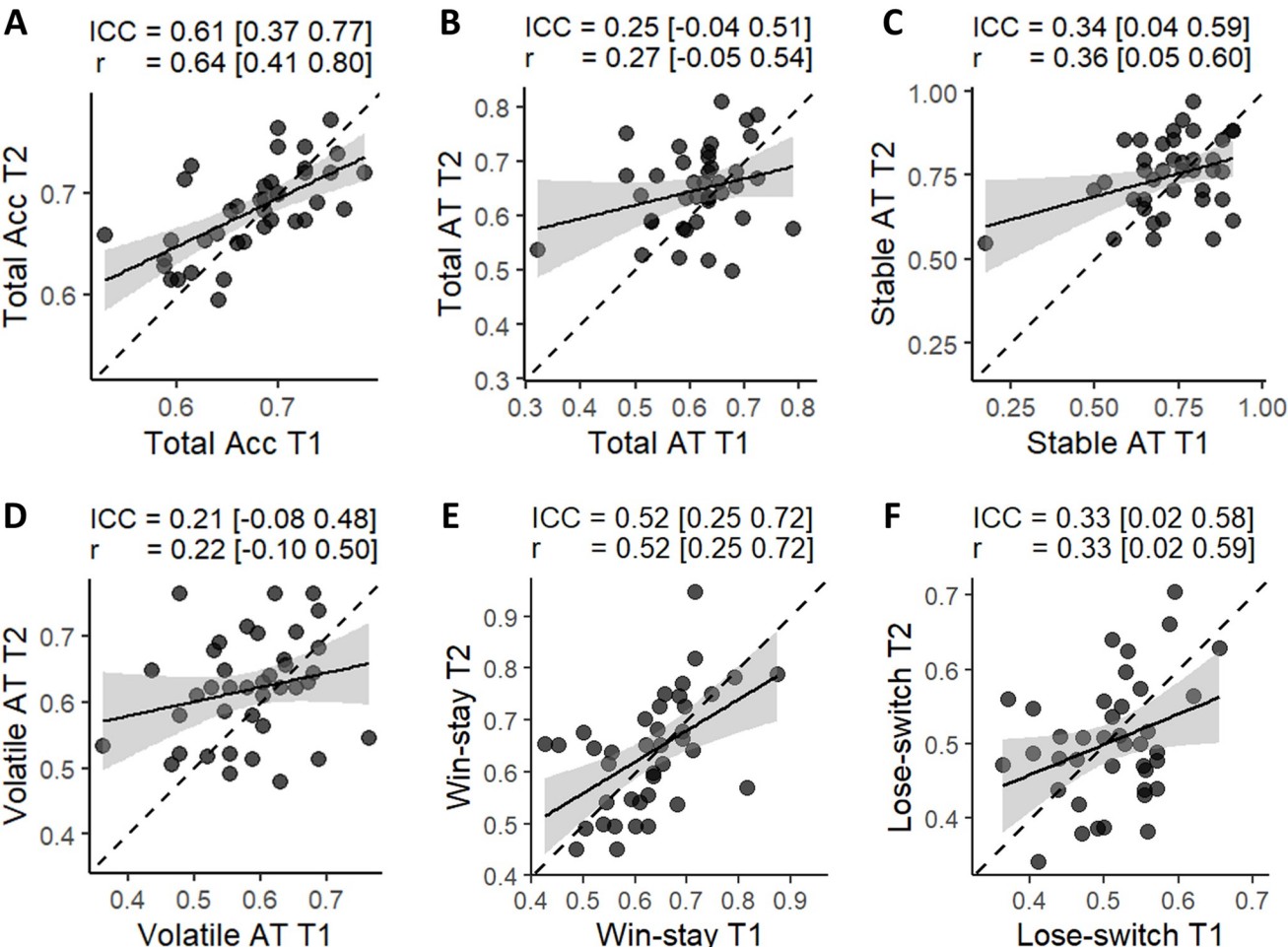

**Fig 3. Test-retest reliability of behavioral measures.** (**A**) Total accuracy. (**B**) Total advice taking. (**C**) Advice taking during the stable phase. (**D**) Advice taking during the volatile phase. (**E**) the probability of staying with the same advice taking strategy (trusting advice or not) after a win in a previous trial (**F**) the probability of changing the advice taking strategy (trusting advice or not) after a loss in a previous trial. ICC(A,1) and Pearson's correlation coefficients are shown above each panel. The square brackets indicate 95% confidence intervals.

## Model recovery and Bayesian model selection

In line with previous work [11], we found that models 1 and 2 are well recoverable (Fig 1 in S1 Appendix). We also found model 2 (mean-reverting HGF) to be the winning model (Fig 2 in S1 Appendix). This was consistent across the two task sessions. Protected exceedance probability for this model was ∼0.7 for session 1 and ∼1 for session 2. Note that < 0.8 might be considered inconclusive, but our results do not hinge on this because we found similar test-retest reliability for both models.

## Test-retest reliability of computational measures

Next, we assessed test-retest reliability of all model parameter estimates of the winning model (Fig 4). Similar to the behavioral measures, we found most parameter estimates to be in the poor reliability range ($ICC < 0.5$), except for decision noise, which was slightly higher ($ICC = 0.56$). The lowest reliability was exhibited by $\kappa_2$ at $ICC = 0.04$. However, further analysis

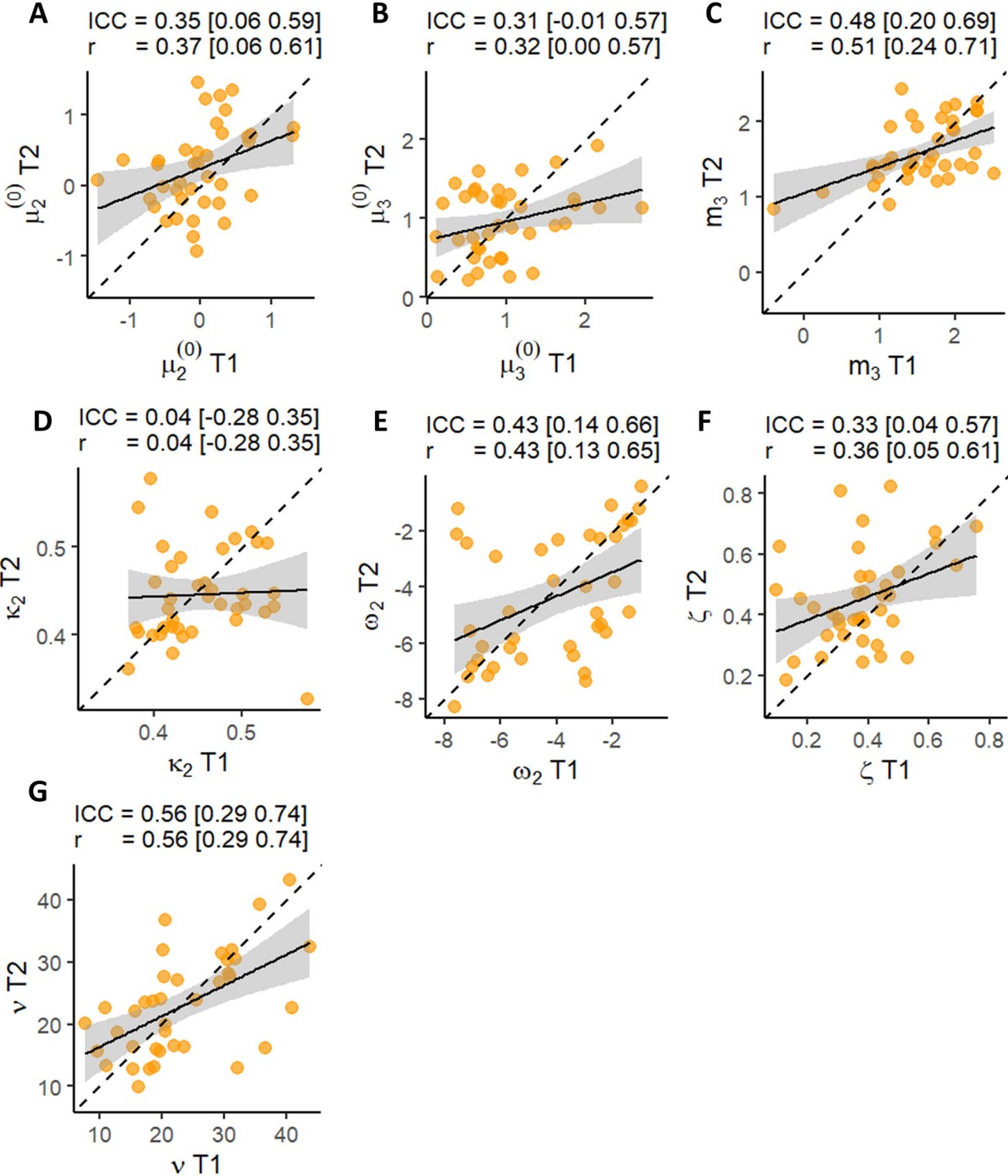

**Fig 4. Test-retest reliability of computational measures.** (**A**) Prior expectations about the adviser's fidelity before starting the task (**B**) Prior expectations about the adviser's volatility before starting the task. (**C**) Volatility equilibrium point (**D**) Phasic learning rate about the adviser's fidelity (**E**) Baseline learning rate about the adviser's fidelity (**F**). The relative weighing of the advice compared to the non-social cue. (**G**) Decision noise. ICC(A,1) and Pearson's correlation coefficients are shown above each panel. The square brackets indicate 95% confidence intervals.

showed that removing data points with outsized effect on ICC as based on Cook's distance drastically improved the reliability of this parameter to $ICC = 0.40$ [0.09 0.64] (see Fig 3 in S1 Appendix for more details). Note, the results for $v$ in Fig 4G exclude one extreme outlier that had a value of $\sim 100$ at T1 (see Fig 3G in S1 Appendix for the outlier).

## Parameter recovery vs test-retest reliability

To investigate to what extent poor test-retest reliability of parameter estimates was due to intrinsic measurement error (as opposed to within-subject variability), we compared test-retest reliability results to parameter recovery results. Parameter recovery was variable across the model parameters, varying from poor to excellent (Fig 5), with three of the model parameters being in the poor recoverability range $ICC \approx 0.5$, and four parameters being within moderate-to-excellent range ($ICC > 0.7$). The most recoverable parameter was $\omega_2$ ($ICC = 0.96$), and the least recoverable parameter was $\kappa_2$ ($ICC = 0.44$).

Test-retest reliability was substantially lower than parameter recoverability for all parameters, but especially for $\zeta$, $\omega_2$, $m_3$, and $\kappa_2$ (Fig 5H). This indicates that for these parameters a major source of variance underlying low test-retest reliability was within-subject variability. These results were very similar when analyzing model 1 as well (Fig 4 in S1 Appendix).

We also investigated whether the low parameter recoverability for some parameters ($\mu_2^{(0)}$, $\mu_3^{(0)}$ and $\kappa_2$) was due to collinearity among parameters. To test this, we fixed these parameters to their prior values one by one and examined if it improved the recovery of the remaining parameters. We found that it did not, suggesting that collinearity was not the reason behind low recoverability (see Fig 5 in S1 Appendix for details).

## Practice effects

Next, we investigated to what extent poor reliability of behavioral and computational measures was due to practice effects (i.e., participants getting better at the task with more practice). We used a paired t-test to test for differences between the two sessions, and we computed Bayesian factors to assess the null hypothesis.

Among the behavioral measures we found significant increases during the second session in advice-taking during the stable phase, close to significant increases in advice-taking during the volatile phase, significant increases in advice-taking overall, and increased win-stay probability (Fig 6). Notably, these changes in task performance did not result in improved performance accuracy ($BF_{01} = 5.6$).

Among the computational measures, only $\mu_2^{(0)}$ and $\zeta$ estimates were significantly different between the sessions (Fig 7). Both of these parameters were found to be increased in the second session, indicating that participants expected the adviser to be more helpful before starting the second session ($\mu_2^{(0)}$), and tended to rely on the social cue more throughout the task ($\zeta$) For all other parameter estimates, there was substantial evidence ($BF_{01} > 3$) for there being no practice effects.

It is interesting to note that while for both $\mu_2^{(0)}$ and $\zeta$, practice effects were of moderate size (Cohen's $d = 0.37$ and $d = 0.42$, respectively), its contribution to low test-retest reliability of these parameters was minimal, reducing it only by 0.02 and 0.03, respectively, as indicated by the differences between the absolute agreement ICC and Pearson's correlation (Fig 4).

To make sure that different neuroimaging environments (EEG vs. fMRI) did not confound practice effects, or test-retest reliability results more generally, we compared parameter estimates between EEG and fMRI sessions. We found substantial evidence ($BF_{01} > 3$) that none of

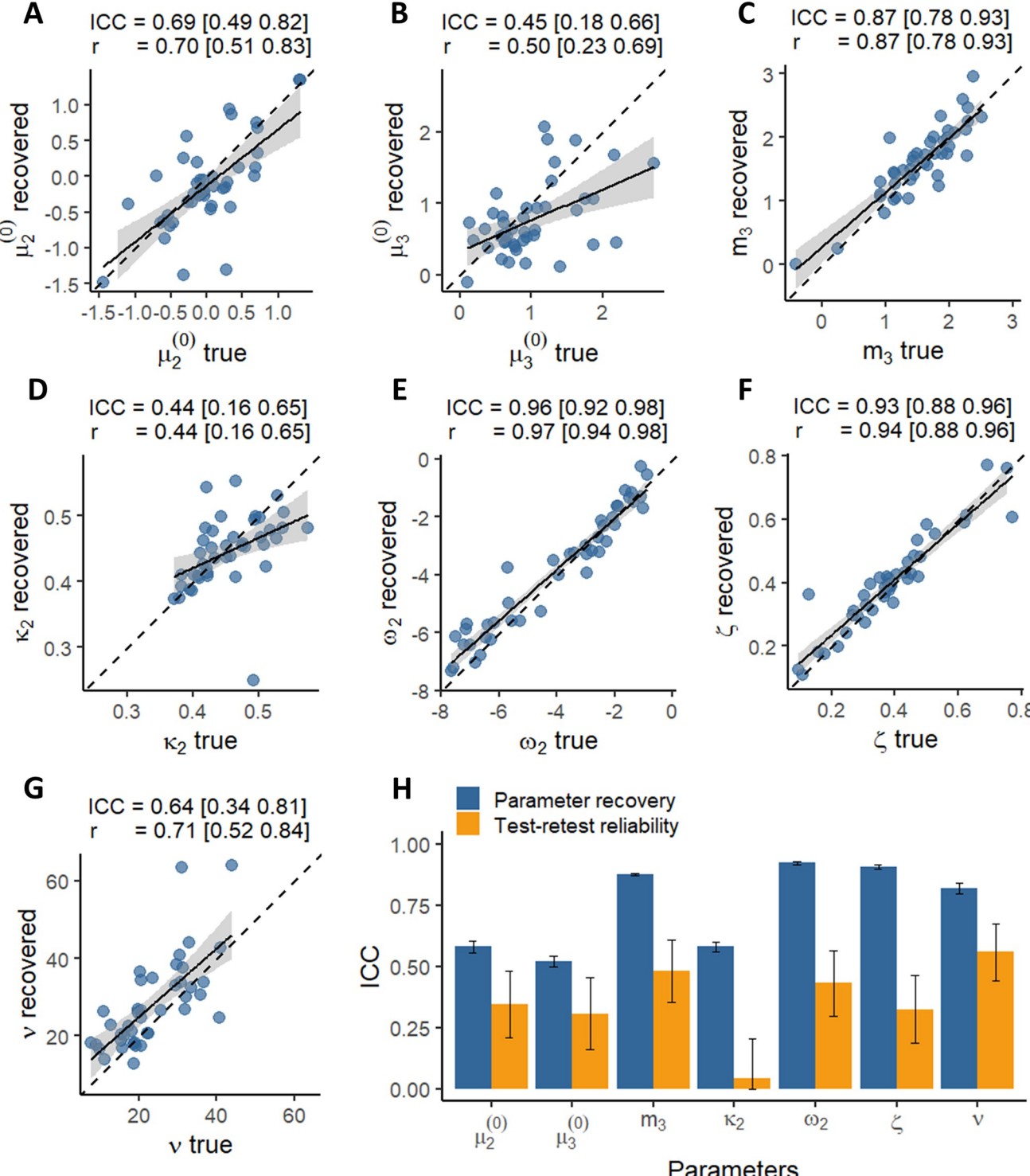

**Fig 5. Parameter recovery and test-retest reliability.** (**A**) Prior expectations about adviser's fidelity before starting the task (**B**) Prior expectations about the adviser's volatility before starting the task. (**C**) Volatility equilibrium point (**D**) Phasic learning rate about the adviser's fidelity (**E**) Baseline learning rate about the adviser's fidelity (**F**). The relative weighing of the advice compared to the non-social cue. (**G**) Decision noise. ICC(A,1) and Pearson's correlation coefficients are shown above each panel. The square brackets indicate 95% confidence intervals. (**H**) Comparison of parameter recovery vs. test-retest reliability for each model parameter. Note, (**A-G**) results are based on one particular random number generator seed (used to generate synthetic data), while (**H**) shows results averaged across 20 simulations with different seeds. The error bars denote standard error.

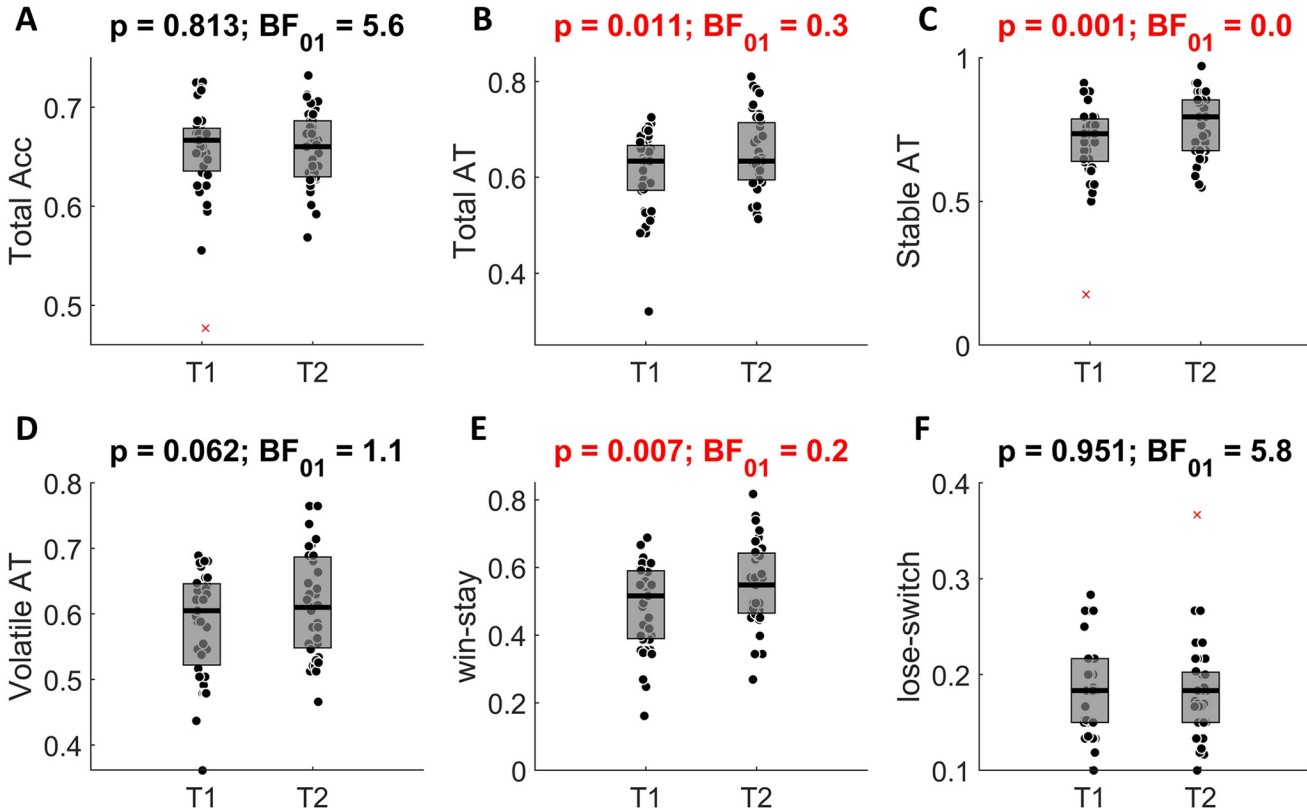

**Fig 6. Practice effects in the behavioral measures.** (**A**) Total accuracy. (**B**) Total advice taking. (**C**) Advice taking during the stable phase. (**D**) Advice taking during the volatile phase. (**E**) the probability of staying with the same advice taking strategy (trusting advice or not) after a win in a previous trial (**F**) the probability of changing the advice taking strategy (trusting advice or not) after a loss in a previous trial. Red crosses indicate outliers. p-values and Bayesian factors for the null hypothesis ($BF_{01}$) of paired t-tests are presented above each plot.

the computational measures were different between EEG and fMRI sessions (Fig 6 in S1 Appendix).

To further investigate how practice effects found in behavioral measures and parameter estimates relate to one another, we performed a correlational analysis (Fig 8). Changes in $\zeta$ were found to be associated with changes in all three behavioral measures, while changes in $\mu_2^{(0)}$ were associated only with changes in advice-taking during the stable phase. These results contribute to face validity of the model (see the following section for further results on this): $\mu_2^{(0)}$ is a prior of a dynamic variable that gets updated throughout the task and would be expected to have a detectable effect only at the beginning of the task, which is what we find (with the stable phase being the first 35 trials). $\zeta$ on the other hand, remains fixed throughout the task, and thus we see its effects on all three measures.

### Face validity of the task and the model

Given the rather poor test-retest reliability results, we performed further analyses to assess to what extent the assay was measuring what it was intended to measure. First, we compared explicit reports of adviser's helpfulness with the actual helpfulness (Fig 9A and 9C), and found that participants consistently reported the adviser to be more helpful when the adviser was indeed more helpful. This indicates face validity of the task—i.e., that the intended manipulation worked.

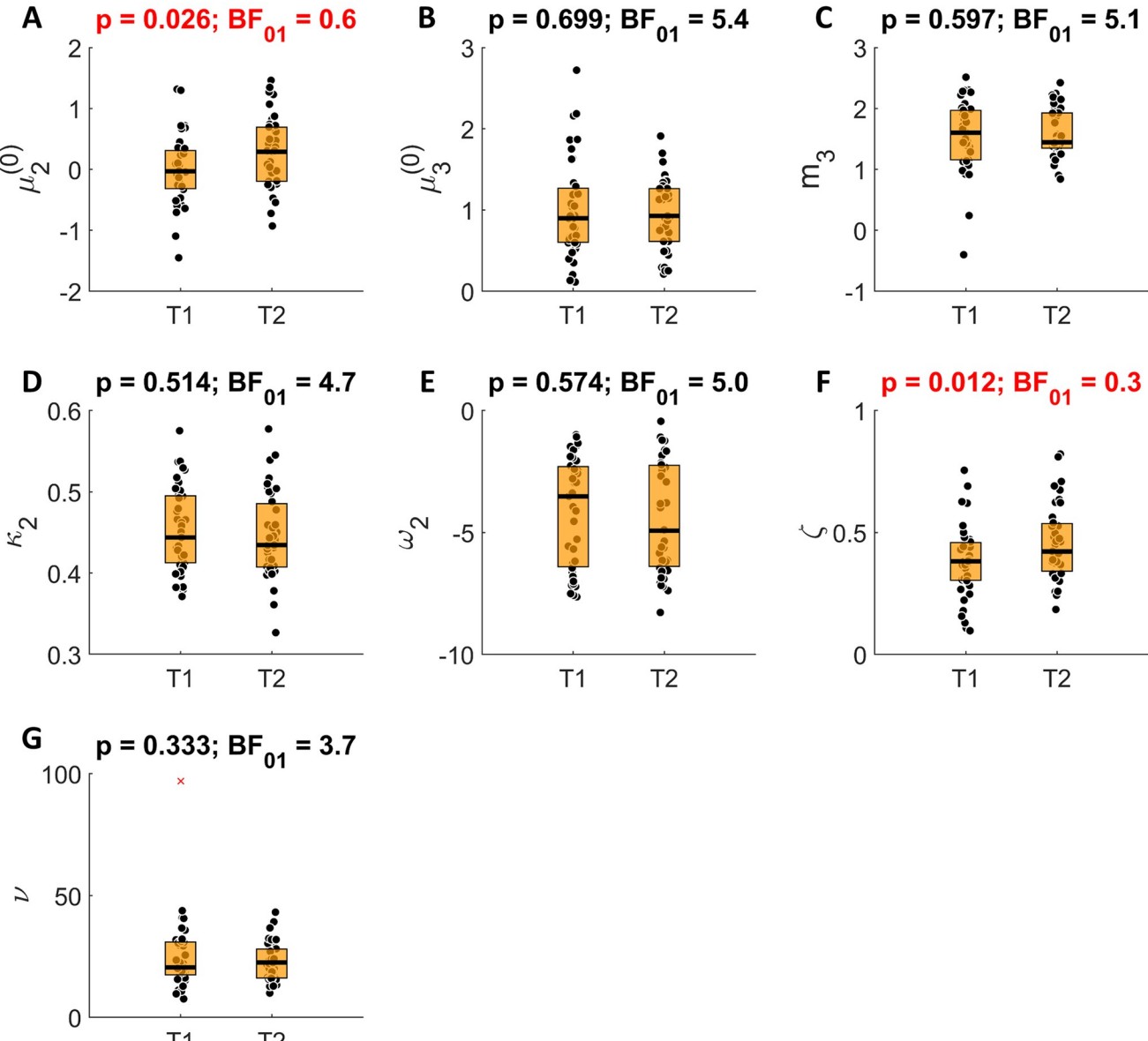

**Fig 7. Practice effects in the computational measures.** (**A**) Prior expectations about adviser's fidelity before starting the task (**B**) Prior expectations about the adviser's volatility before starting the task. (**C**) Volatility equilibrium point (**D**) Phasic learning rate about the adviser's fidelity (**E**) Baseline learning rate about the adviser's fidelity (**F**). The relative weighing of the advice compared to the non-social cue. (**G**) Decision noise. Red crosses indicate outliers. p-values and Bayesian factors for the null hypothesis ($BF_{01}$) of paired t-tests are presented above each plot.

Next, we investigated whether model-derived estimates of inferred adviser's fidelity ($\hat{\mu}_2$) agreed with explicitly reported inferred fidelity at different points in the task (Fig 9B and 9D). We found that $\hat{\mu}_2$ was significantly higher when advice was reported to be helpful vs when it was reported misleading ($p < 0.01$ in both testing sessions). This suggests that $\hat{\mu}_2$ captures what it is assumed to capture, which contributes to the face validity of the model.

## Discussion

Here, we assessed psychometric properties of a computational assay designed to probe how people infer intentions in the context of receiving volatile advice [10]. We found this assay to

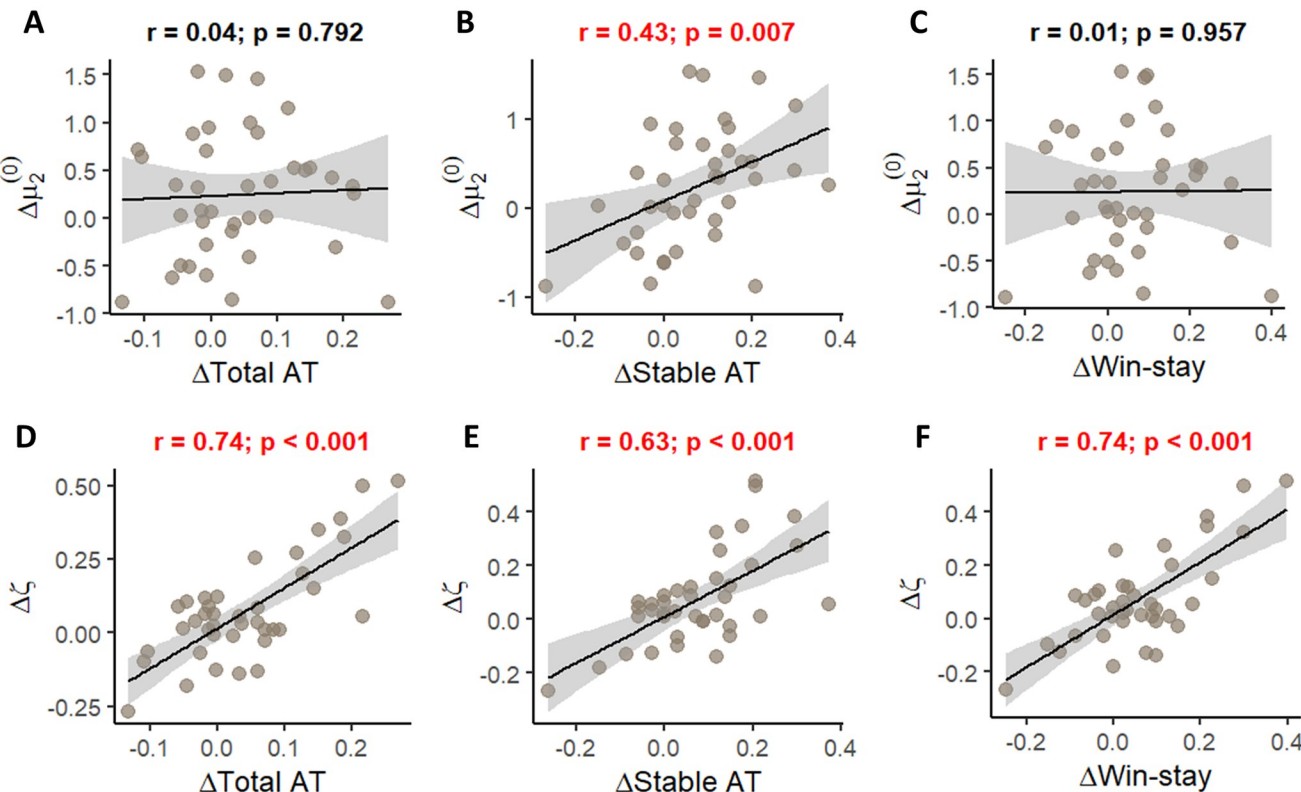

**Fig 8. Correlation between the changes in behavioral and computational measures.** (**A-C**) Prior expectations about adviser's fidelity before starting the task vs. total advice taking (**A**), vs. advice taking during the stable phase (**B**), and vs. win-stay frequency (**C**). (**D-F**) The relative weighing of the advice compared to the non-social cue vs. total advice taking (**D**), vs. advice taking during the stable phase (**E**), and vs. win-stay frequency (**F**). Pearson's correlation coefficients and corresponding p-values are presented above each panel.

have mostly poor test-retest reliability: this was true for both behavioral and computational measures. Our work is one of the first one to investigate test-retest reliability of a HGF model (see [32]), and it adds to the emerging body of research showing that many computational assays used to study mental disorders suffer from poor psychometric properties, which poses considerable challenges for their use in research and for clinical translation efforts [9].

## Why is test-retest reliability low?

Although many studies report poor test-retest reliability of computational measures [9], post hoc analysis of different sources of variance affecting reliability is seldom performed. Here, we sought to tease apart the underlying sources of variance by directly comparing parameter recoverability with test-retest reliability for each parameter. Parameter recovery is constrained only by measurement error (and is not affected by within-subject variability across testing time points) providing an upper bound on test-retest reliability [9]. In our study, measurement error proved to be an important factor behind low test-retest reliability: three parameters had poor recoverability (in the range of $ICC \approx 0.5$), indicating high measurement error for these parameters.

Still, for most parameters test-retest reliability was substantially lower than parameter recoverability suggesting that within-subject variance also played a major role in reducing test-retest reliability. We considered three sources of within-subject variance: practice effects, state-

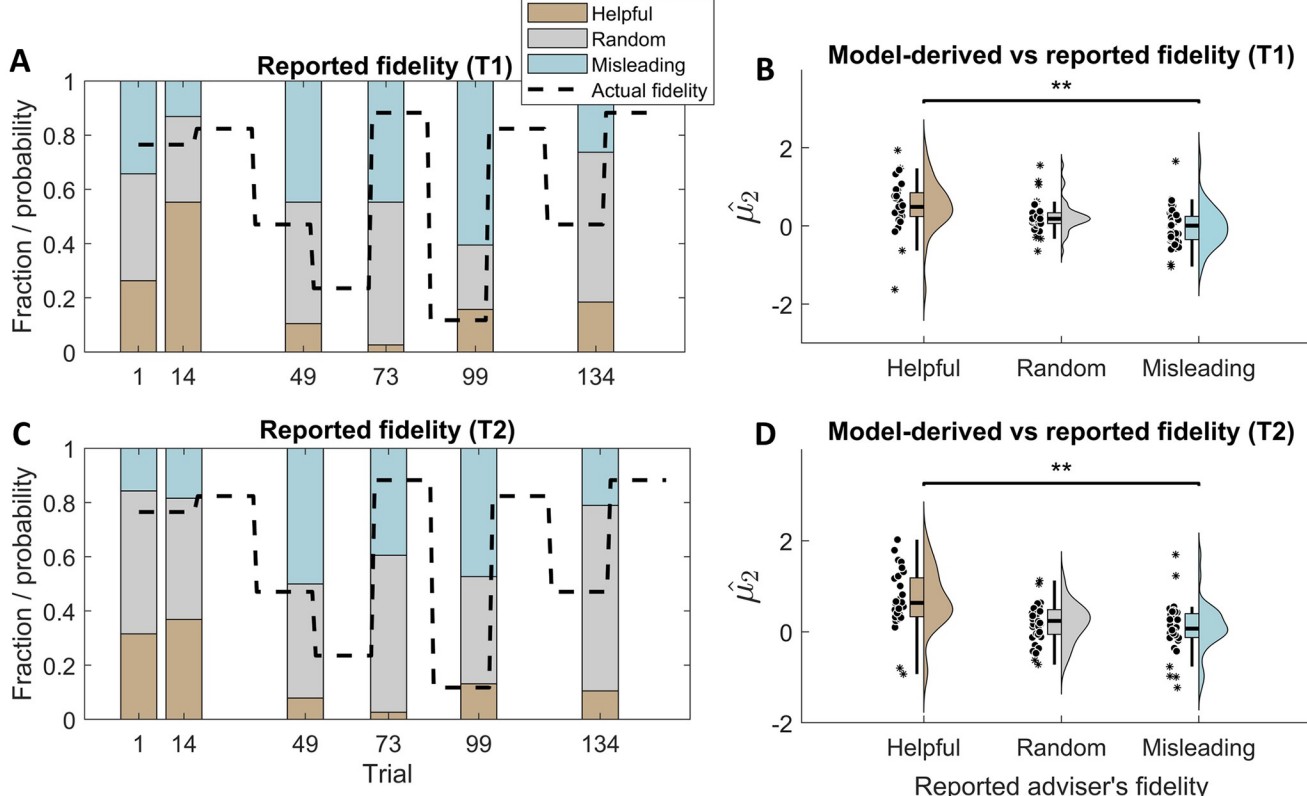

**Fig 9. Face validity: The task and the model.** Reported adviser's fidelity vs actual fidelity during test (**A**) and retest (**C**) at different points in the task. Model-derived estimates of adviser's fidelity ($\hat{\mu}_2$) broken down according to the explicitly reported fidelity at the corresponding points in the task during test (**B**) and retest (**D**).

like fluctuations, or trait-like changes. We did find significant practice effects in some parameters, and while these effects were of moderate size (Cohen's $d = 0.37 - 0.42$), they accounted only for a small fraction ($ICC \approx 0.02 - 0.03$) of the reduction in test-retest reliability. Together with other studies reporting small to medium practice effects [33, 34], it suggests that fixed practice effects might be a fairly minor factor behind low test-retest reliabilities of computational assays (although see [35] for a more extensive analysis). Nonetheless, practice effects still remain understudied and under-reported [9].

Our test-retest interval was short (2 weeks), which rules out the possibility that trait-like changes (which by definition happen at longer timescales) affected the observed test-retest reliability. This means that the majority of within-subject variance might be explained by state-like fluctuations (mood, motivation, energy levels, etc.). However, since we did not collect such data, we are unable to demonstrate this definitively and to examine which state-like factors may be driving the effects.

In general, the degree to which state-like fluctuations impact computational measures remains understudied, even though previous work indicates its importance [36–38]. Studies investigating these effects in the context of test-retest reliability are only starting to emerge: Sullivan et al. [39] found current mood intensity (positive or negative) to be associated with learning rate estimates in the Iowa gambling task, Schaaf et al. [40] found stress and happiness levels to be associated with the loss learning rate parameter in Two-armed Bandit task, while

Schurr et al. [35] found affective valence to be associated with the inverse temperature parameter in Intertemporal Choice task, the lapse rate in Go/No-Go task, and the weight of relative uncertainty in Two-armed Bandit task. However, the effect sizes in all of these studies were rather small. Furthermore, Schurr et al. [35] explicitly showed that valence explained only a small fraction of within-subject variance in parameter estimates while most of the variance was explained by subject-specific noise terms (the noise term simply captured any remaining unexplained variance). This indicates there could be many more state-like factors that might be driving these effects and need to be further examined.

Another recent study found that sleep deprivation affected perceptual stability in random dot kinematogram task, meanwhile psychosis proneness (a schizotypal trait) did not have any effect on task performance [41]. While this study looked at behavioral rather than computational measures, it nicely highlights that in some cases, state-like factors may have stronger effects than the primary variables of interest (e.g., traits, symptoms, diagnostic categories, etc.).

Overall, our work highlights that by directly comparing recoverability with reliability one can begin teasing apart which sources of variance are responsible for low test-retest reliability. While a handful of studies investigating test-retest reliability of computational measures have reported parameter recovery results [33, 34, 39, 42–46], such direct comparison has not been performed. Pooling the data from these studies does show that test-retest reliability tends to be substantially lower than parameter recoverability, highlighting the importance of accounting for within-subject variability (Fig 7 in S1 Appendix).

It must be noted, however, that our attempt to tease apart the different sources of variance has some caveats. First, the parameter recovery analysis only indicates the intrinsic measurement error, while the actual measurement error is likely to be higher. That is, given that we cannot assume the model to perfectly match the underlying cognitive mechanisms (which is true by construction, because modelling in its nature is an attempt to simplify reality), the mismatch will introduce additional measurement error. Second, our practice effects analysis accounts only for fixed effects. Any kind of idiosyncratic practice effects would be statistically indistinguishable from measurement error. Thus, it is possible that in addition to the state-like factors, that we emphasize in our interpretation, the remaining within-subject variance might also be driven by idiosyncratic task strategies and heuristics. However, how such strategies are selected may still be affected by state-like factors—ultimately blurring the distinction between state-like effects and idiosyncratic practice effects. Our ability to explore these and other more nuanced questions was limited by our experimental design, but should be investigated in the future using more intentional experimental designs (e.g., see [35]).

Additionally, the social nature of the task might have led participants to employ higher-level strategies during their first session based on prior expectations of potential deception. Although the advice-taking task did not involve deception and participants were truthfully informed that the adviser had a different incentive structure, the lack of explicit details about this structure until debriefing might have fostered skepticism towards the task and experimenter's instructions. Potential concerns of deception may have influenced participants' behavior uniquely during the first session. By the second session, participants would have become familiar with the procedure, potentially reducing their concerns about deception. Consequently, the first session may differ substantially from subsequent sessions: we already saw this reflected in practice effects at the group level, but the individual differences in these effects would be adding to error variance. Therefore, to obtain more accurate insights into test-retest reliability, future studies should consider examining the reliability between the second and third sessions, after participants' initial concerns about deception have been alleviated.

### What are the implications of our findings for the previous work using this assay?

The only previous work that used the exact version of the assay, the same modelling pipeline, and some of the same data was done by Hauke et al. [11], which examined first-episode psychosis patients, individuals at clinical high-risk for psychosis, and matched healthy controls (note that this study had slightly less trials, 136 instead of 153 analyzed here; however, this had a negligible effect on reliability—see Fig 8 in S1 Appendix). Hauke et al. [11] found that computational measures ($m_3$ and $\kappa_2$) to be different across the groups as well as showing weak correlations with clinical symptoms—although these results did not survive correction for multiple comparisons. Our finding of $\kappa_2$ having a near-zero test-retest reliability makes it even less likely that the group differences in $\kappa_2$ reported in Hauke et al. [11] are true positives. However, our further analysis suggested that $\kappa_2$ reliability may have been slightly underestimated due to an outsized effect of 4 participants, with the ICC being $\sim 0.4$ after removing them (Fig 3D in S1 Appendix). In such case, as well as for $m_3$, which also showed similar reliability of ICC $\approx 0.5$, the group differences are less likely to be in question—while this level of reliability significantly reduces the observable effect size (e.g., [47, 48]), if the true effect is very large (Cohen's $d > 1$) it would still be detectable with the sample size that the study had.

When it comes to other studies using this assay, our findings are less straightforward to generalize because each study used a slightly different version of the assay. For example, some previous studies used different volatility manipulations, used a different version of HGF and had either fewer (120) [14] or more (210) trials [49, 50]. On top of that, some other studies used a more difficult version of the task where instead of being shown explicit reward probabilities in a form of pie chart on every trial participants had to infer these reward probabilities from feedback [12, 13]. All of these differences among different versions of the task can lead to varying degrees of test-retest reliability [9, 51]. This stresses the importance of making test-retest reliability analysis a routine practice [9, 52].

### How to improve test-retest reliability?

Recent work has demonstrated that modelling task performance jointly across the two testing sessions can improve reliability [39, 42, 53]. In this approach, each parameter is estimated in a form of a bivariate distribution, from which a correlation coefficient between the sessions can be derived. A simulation analysis by Waltmann et al. [42] has demonstrated that estimating reliability this way is more accurate compared to modelling the sessions separately. However, computing ICCs using parameter estimates obtained from the joint modelling of the sessions resulted in inflated reliability [42]. It means that using these point estimates in further analyses without accounting for their uncertainty and their non-independence would also introduce biases. To avoid these issues, future analyses should use full posterior distributions rather than relying solely on point estimates. Nonetheless, here we used a simpler modelling approach that relies on fixed priors (see Methods for details), as this is the only established optimization pipeline available for fitting HGF models [17]. Joint modelling of the sessions—or even modelling the sessions separately but with empirical priors—poses significant challenges for the HGF. Without fixed priors, there is less control over regularization, increasing the risk of parameter values falling into regions that violate model assumptions and causing the optimization to fail. These limitations will be addressed in future work.

While hierarchical modelling approaches may provide some improvement in the reliability of parameter estimates, focusing on task design optimization seems to be a much more promising avenue as it can address the root causes of low reliability. Most cognitive tasks have been optimized for producing strong group effects, which often means low between-subject

variation and as a result, low test-retest reliability [54]. A new wave of research is needed that places more emphasis on high test-retest reliability: see [51] for some of the main task design factors to consider.

The first thing to ensure in task optimization is high parameter recoverability. Here we found that some parameters were recovered poorly, making it impossible to have high test-retest reliability for these parameters. Note that parameter recoverability of $\sim 0.5$ would not have been considered too problematic in previous research, as it would be interpreted as a 'large' effect size. However, considering parameter recoverability as the upper limit for test-retest reliability [9] imposes much stricter criteria for what recoverability can be considered satisfactory.

How the parameter recoverability itself could be improved does not have a straightforward answer. Sometimes low recoverability of parameters stems from collinearity—when parameters trade-off among each other to provide an equally good fit to the data. Here we ruled out this possibility by fixing poorly recovered parameters one by one and showing that it did not improve recoverability of the remaining parameters (Fig 5 in S1 Appendix). Another potential way to improve parameter recovery is to increase the number of trials. However, because our task depends on trial-by-trial dynamics, it immediately reverts us back to a more general question of what advice probability structure should be chosen (e.g., how should the volatility of the adviser change over time). It is likely that a different structure than was used in this task could constrain the parameter estimates better—however, discovering such structure in a principled way is a rather challenging problem. Finally, one more easily addressable factor that may be limiting parameter recoverability is the binary nature of responses in this task [32]. Future studies should consider incorporating continuous response measures, which contain more information and can therefore constrain parameter estimates better.

## Construct validity

Despite the low test-retest reliability, we were able to show that the task has face validity at the group level: (1) participants reported the adviser to be more helpful when the adviser was indeed more helpful (task face validity) and (2) model-derived estimates of inferred adviser's fidelity ($\hat{\mu}_2$) covaried with explicitly reported inferred adviser's fidelity (model face validity). This adds to previous validation work that also found HGF parameters in this task to be associated with interpersonal reactivity index (IRI) scores [10].

A more thorough validation of the assay would require demonstrating convergent and divergent validity of each model parameter across a battery of tasks; see [55] for an example in the context of reinforcement learning models and [56] for an example in the context of drift-diffusion models. Ideally, these tasks would vary in their similarity among multiple important dimensions (e.g., instructions, stimulus types, volatility protocols, adviser qualities, etc.), so that a sufficient variety of tasks is achieved for assessing convergent and divergent validity. This work could be carried out in parallel with testing the predictive and longitudinal validity of these measures [9].

## Limitations

Our sample size in this study was relatively small (N = 39), which resulted in rather wide confidence intervals around the estimated mean ICC values. While there is no consensus for the optimal sample size for reliability studies—the suggested optimal sample sizes range from $\sim 60$ [57], to 100 [58], to 400 subjects [59]—larger samples can provide more precise reliability estimates, which is especially important when reliability itself is poor. Another limitation in our study is the fact that our test and retest sessions were done during EEG and fMRI

scanning. Even though we showed that there were no fixed effects on task performance between the two experimental environments, it may have nonetheless contributed to the error variance.

## Conclusion

Our findings of the largely poor test-retest reliability of behavioral and computational measures of advice taking under volatility warrant more caution when interpreting previous findings of this [11] and potentially other versions of the assay [12–14]. More generally, our findings add to the growing body of research showing that many behavioral and computational task measures suffer from poor test-retest reliability [9, 54, 60]. We sought to illustrate how using simple methods, one can start to disentangle the different sources of variance underlying low test-retest reliability of computational measures, however, more intentional experimental designs are needed for a more in-depth analysis of these sources (e.g., [35]). A better understanding of what factors drive low test-retest reliability would facilitate the design of more psychometrically sound assays, which would increase the quality of future research and the possibility of clinical translation.

## Supporting information

**S1 Appendix. All supplementary materials.**
(PDF)

## Author Contributions

**Conceptualization:** Povilas Karvelis, Andreea O. Diaconescu.

**Data curation:** Daniel J. Hauke.

**Formal analysis:** Povilas Karvelis, Daniel J. Hauke.

**Funding acquisition:** Andreea O. Diaconescu.

**Investigation:** Andreea O. Diaconescu.

**Methodology:** Povilas Karvelis, Daniel J. Hauke, Andreea O. Diaconescu.

**Project administration:** Michelle Wobmann, Christina Andreou, Amatya Mackintosh, Renate de Bock, Stefan Borgwardt, Andreea O. Diaconescu.

**Resources:** Michelle Wobmann, Christina Andreou, Renate de Bock, Stefan Borgwardt.

**Supervision:** Andreea O. Diaconescu.

**Visualization:** Povilas Karvelis.

**Writing – original draft:** Povilas Karvelis.

**Writing – review & editing:** Povilas Karvelis, Daniel J. Hauke, Christina Andreou, Renate de Bock, Andreea O. Diaconescu.

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
