## [Decision Letter · Decision Letter 0]

26 Apr 2024

PONE-D-23-38672Test-retest reliability of behavioral and computational measures of advice taking under volatilityPLOS ONE

Dear Dr. Karvelis,

Thank you for submitting your manuscript to PLOS ONE. After careful consideration, we feel that it has merit but does not fully meet PLOS ONE’s publication criteria as it currently stands. Therefore, we invite you to submit a revised version of the manuscript that addresses the points raised during the review process.

Please respond all comments and highlight the changes.

We look forward to receiving your revised manuscript.

Kind regards,

Thiago P. Fernandes, PhD

Academic Editor

PLOS ONE

Journal Requirements:

2. Please expand the acronym “CIHR” (as indicated in your financial disclosure) so that it states the name of your funders in full.

3. Thank you for stating the following in the Acknowledgments Section of your manuscript: "PK is supported by Canadian Institutes of Health Research (CIHR) Fellowship (472369). AOD is supported by Swiss National Science Foundation and the Krembil Foundation (1000824). The funders played no role in the study design, data collection and analysis, decision to publish, or preparation of the manuscript."

Please remove any funding-related text from the manuscript and let us know how you would like to update your Funding Statement. Currently, your Funding Statement reads as follows: "PK is supported by Canadian Institutes of Health Research (CIHR) Fellowship (472369). AOD is supported by Swiss National Science Foundation and the Krembil Foundation (1000824). The funders played no role in the study design, data collection and analysis, decision to publish, or preparation of the manuscript."

Reviewers' comments:

Reviewer's Responses to Questions

**Comments to the Author**

1. Is the manuscript technically sound, and do the data support the conclusions?

Reviewer #1: Yes

Reviewer #2: Partly

2. Has the statistical analysis been performed appropriately and rigorously? 

Reviewer #1: Yes

Reviewer #2: Yes

3. Have the authors made all data underlying the findings in their manuscript fully available?

Reviewer #1: Yes

Reviewer #2: Yes

4. Is the manuscript presented in an intelligible fashion and written in standard English?

Reviewer #1: Yes

Reviewer #2: Yes

5. Review Comments to the Author

Reviewer #1: The paper covers a topic of significant importance and had been written very well. The extensive statistical analysis and depth covered by statistical descriptions provides for an interesting and informative read. Even though, based on my level of understanding, there were no technical issues that require corrections, addressing the below noted possible writing discrepancies would make the paper even better:

• Page 2, Line 68: Would it be more appropriate to start the sentence with a word rather than a numerical text? Four as opposed to 4.

• Page 2: It appears that the task of the participants involves a total of 153 trials. The text of the methodology section does not seem to specifically mention this. However, it could be inferred from details from different parts of the text (e.g. page 6) as well as Figure 1B. It would be better if this was specifically noted under section 2.3.

• Page 2, Line 94: The word ‘they’ seem to be repeated.

• Page 4, Equation 4: It seems the ‘x’ shown as a subscript should be ‘3’.

• Page 4, Equation 5: Again, it seems the ‘x’ shown as a subscript should be ‘3’.

• Page 5, Line 150: I wonder if the two models are named in the wrong order, considering how they are discussed later in the paper? Shouldn’t Eq. 5 HGF with the mean reverting process be model 2, and Eq. 4 HGF without the mean reverting process be model 1?

• Page 11: It seems Figure 7 is not referred to in the text. Perhaps it was meant to be noted at Page 10, Line 263?

• Page 11, Line 282: Should the “effects an all three” be “effects on all three”?

• Page 12, Line 288: There is a repeated ‘validity’ term.

Reviewer #2: This article presents evidence from N=39 subjects that the behavioural and computational measures of a task assessing advice taking under volatility have generally poor test-retest reliability, partially due to poor parameter recoverability. Given that the task and computational model have already been used to study psychiatric disorders and development, the article is relevant and timely, contributing to a growing body of research documenting the reliability of task-derived measures of interindividual differences. Clear merit notwithstanding, there are a few key points that I believe the authors should address before the manuscript is ready for publication.

Major:

1. Methods:

a. Participants: Is there any information on age, gender, SES or other demographics / sample characteristics? If so, this should be provided. Also, it does not become clear whether the participants were healthy controls, FEP patients, or people at high risk for psychosis.

b. Behavioural measures: There is no description as to how behavioural measures are derived. Even if this is done by simply taking means, this should be described properly. I would also strongly encourage the use of trial-by-trial analyses using hierarchical models (see Rouder & Haaf’s 2019 paper "A psychometrics of individual differences in experimental tasks"). I understand from the discussion that the authors have qualms here – please see below for my response to those qualms.

c. Computational model: The authors are very technical in their description of the modeling approach. I think it is important to motivate the use of the model – and indeed the task itself – at a higher level. Relevant questions to answer are: What does the task measure and why do we care? Is there an example of where we might find its metrics useful to better describe/understand a disorder or similar? What do the different behavioural metrics tell us? What do the different computational parameters tell us? How do they relate to behaviour (on a theoretical level, but also maybe empirically)? (Some of this could go to the introduction). The technical description could also be improved. Not every variable in the model is properly explained (I cannot find an explanation of z, for example, or s). Likewise, technical descriptors should be defined (e.g., what is a tonic learning rate in this context? What is coupling strength?)

d. Model space and model selection: It would be good to separate model space and model selection (which should come after model fitting?). The authors should also make sure they explain all acronyms (TAPAS, VBA, mHGF, sHGF). Further, it would help if the authors were a bit clearer in their prose. For example, what does it mean for a model to be “implemented using ‘tapas_hgf_binary’ function” (l.156-7)? What does the function do? Likewise, l. 159, what two hypotheses do the authors mean? They could either be stated explicitly, or the phrasing changed to “to arbitrate between the two models”.

e. Model fitting: The authors do not properly motivate why they fixed the meta-volatility and drift-rate parameters. Is there a conceptual reason (are those parameters not of interest? If so, why not?), or perhaps an empirical reason (models fail to converge)? The fitting procedure itself also deserves more attention. What is it that is optimised using the quasi-Newton optimisation algorithm? I assume a posterior likelihood, given that there are priors involved? This should be stated. The priors themselves should be presented in the main body of the article – priors can have important consequences for reliability. Did the authors consider using multivariate priors (i.e., priors that include correlations between parameters?)? If not, why not?

2. Results

a. Parameter recovery vs test-retest reliability:

i. The authors imply that the ICC between true and recovered parameters reflects (inverse) measurement error. My understanding is that parameter recovery primarily assesses a model's capacity to accurately reflect behaviour. That is, whether a specific set of parameter values uniquely maps onto a specific behavioural pattern or whether this pattern can also be achieved using a different set of parameter values. This can be understood to reflect (inverse) measurement error insofar as the model parameters are taken as instruments measuring behaviour? Is that what the authors mean? It would be great if they could provide further clarification on this point.

ii. I wonder if the authors have considered comparing the test-retest reliability to an internal consistency measure rather than (or better, in addition to) the parameter recoverability? The difference between the two should relatively straightforwardly map onto state changes, right?

iii. That said, I think the fact that some model parameters recover poorly receives too little attention (both in the results and the discussion). As the authors say, it provides an upper bound to their reliability (both internal consistency and test-retest) and is therefore absolutely crucial for reliability. Was the recoverability of those parameters also poor in previous applications of the task+model? Is this because some of the parameters trade off against each other? Could this be remedied by (a) finding the optimal number of free parameters in a more extensive model selection procedure, and/or (b) using a multivariate prior that includes correlations between parameters? If not, is there an alternative model that may have better recovery properties? If not, do we have to conclude that this task+model is unsuitable for interindividual difference research altogether?

b. Practice effects: Please explain, for each measure, what an increase or decrease means on a psychological level and why.

c. Face validity: Have the authors considered moving this section to the beginning of the results section? This would also provide an opportunity to explain the model and its parameters at a higher level and in relation to task behaviour.

3. Discussion:

a. Improving test-retest reliability: I believe the authors may have partially misunderstood the analyses by Brown et al. / Waltmann et al. The parameter estimates derived from jointly modelling data from the two sessions should actually be more accurate than those derived by separate modelling of the data. This should be true even though (in fact because) partial pooling biases the estimates towards the within subject and between subject means. Perhaps counterintuitively, the estimates are, on average, closer to the ground truth than unbiased estimates (i.e., estimates that do not use partial pooling across sessions or individuals) (see Effron and Morris’ 1979 paper "Stein’s Paradox in Statistics" for a detailed explanation on how and why this is possible; also see Rouder & Haaf for a more contemporary treatment). As the authors correctly point out, an issue arises when one tries to calculate ICCs from these estimates – these will be inflated. Indeed, calculating ICCs from the estimates may violate the assumptions of the ICC because due to partial pooling, the estimates are no longer independent from one another (this messes with the variances – although, as the authors also state, Brown et al. and Waltmann et al. found a way to circumvent this problem). However, whether the ICCs (i.e., the reliability metrics) are biased or not has no bearing on the accuracy of the parameter estimates themselves – per Stein’s Paradox, they should still be better.

b. Validity: As in the results section, I would move this to the top of the discussion.

c. Limitations: The sample size is very small. I understand that this is because the data is derived from a clinical study and I am glad the authors still put together this paper but it should be clearly stated as a limitation. The same goes for the difference in the context in which the task was performed (EEG vs fMRI) – this is a source of variance that would be controlled in a more targeted reliability study. Even if the authors show that there is no bias (no directed, fixed effect) the change in context is almost certain to introduce (nonspecific) error variance.

d. Conclusion: This could be more task (and perhaps even parameter) specific – what did the authors find and what does it mean for using the task in the future?

Minor:

1. The abstract could be more concrete/specific and more assertive in tone.

2. In the beginning of the introduction, the authors suggest that there is growing enthusiasm that computational measures may become a new basis for psychiatric nosology. Given that no interventions (that I know of) have flown from almost two decades of computational psychiatry research, and (to my knowledge) no unequivocal mappings of computational parameter alterations to symptoms have emerged, I think this is a bit of an overstatement – I rather feel like the field is waking up to its limitations (such as poor reliability)!

3. L. 48 ff: I would suggest moving the periods and commas around to make this more effective: “Up until recently, the psychometric properties of computational measures have received little attention (Browning et al., 2020). Emerging empirical evidence indicates that both reliability and construct validity are often poor (Karvelis et al., 2023) but the psychometric properties of many currently used computational assays remain unknown...”

4. L. 230: “<¡0.8”, is this a typo?

5. L. 252: Do the authors mean “i.e.,” rather than “e.g.,”?

6. L.256-7: Do the authors mean an increase across sessions? It’s best to be precise here.

7. L. 344 ff: This paragraph comes a little out of the blue – can it be integrated with the previous paragraph?

8. L.391: Citation is missing brackets.

6. PLOS authors have the option to publish the peer review history of their article (what does this mean?). If published, this will include your full peer review and any attached files.

Reviewer #1: No

Reviewer #2: No

---

## [Author Response · Author response to Decision Letter 0]

29 Jul 2024

We thank the reviewers for providing insightful comments! The responses are included in the .doc attached to this submission.

---

## [Decision Letter · Decision Letter 1]

20 Aug 2024

PONE-D-23-38672R1Test-retest reliability of behavioral and computational measures of advice taking under volatilityPLOS ONE

Dear Dr. Karvelis,

Thank you for submitting your manuscript to PLOS ONE. After careful consideration, we feel that it has merit but does not fully meet PLOS ONE’s publication criteria as it currently stands. Therefore, we invite you to submit a revised version of the manuscript that addresses the points raised during the review process.

Thank you for your valuable submission. The authors will note that some concerns were raised.Please respond to all of them and highlight the changes in the manuscript.

We look forward to receiving your revised manuscript.

Kind regards,

Thiago P. Fernandes, PhD

Academic Editor

PLOS ONE

Journal Requirements:

Reviewers' comments:

Reviewer's Responses to Questions

**Comments to the Author**

1. If the authors have adequately addressed your comments raised in a previous round of review and you feel that this manuscript is now acceptable for publication, you may indicate that here to bypass the “Comments to the Author” section, enter your conflict of interest statement in the “Confidential to Editor” section, and submit your "Accept" recommendation.

Reviewer #2: (No Response)

2. Is the manuscript technically sound, and do the data support the conclusions?

Reviewer #2: Yes

3. Has the statistical analysis been performed appropriately and rigorously? 

Reviewer #2: Yes

4. Have the authors made all data underlying the findings in their manuscript fully available?

Reviewer #2: Yes

5. Is the manuscript presented in an intelligible fashion and written in standard English?

Reviewer #2: Yes

6. Review Comments to the Author

Reviewer #2: The authors have answered almost all of my queries satisfactorily and I congratulate them on their hard work. I particularly appreciate their higher level explanations that make the manuscript much more accessible. I only have two minor queries regarding their responses.

2.a.i. Regarding internal consistency, I understand that it is hard to separate odd and even trials in a learning task, but it is possible to compare parameters derived from the first and second halves of a task. I know that this has its own limitations but I think it might be helpful to at least get a rough idea of how much variance between sessions might be due to state changes vs noise.

3.a. As I mentioned in my previous comment, point estimates derived from hierarchical models are, on average, more accurate than those from flat models (literature refs as before). The fact that they are not independent from one another does not explain, by itself, why using them in further analyses should introduce bias, and in what way or what direction. It would be good if the authors could explain their reasoning in more detail here. Moreover, even if using these estimates did introduce bias, "accounting for their uncertainty and their non-independence" in further analyses as the authors suggest is not difficult – it can be done by simply including the additional variables of interest in the model – and therefore should not, in principle, be too much to ask if hierarchical modelling conveys a significant advantage in estimation accuracy. Is there any particular use case that the authors envision in which this would be unreasonable or impossible?

7. PLOS authors have the option to publish the peer review history of their article (what does this mean?). If published, this will include your full peer review and any attached files.

Reviewer #2: No

---

## [Author Response · Author response to Decision Letter 1]

30 Aug 2024

We would like to thank the reviewer for the chance of providing further clarifications - we believe it helped improve the manuscript.

---

## [Editor Report · Decision Letter 2]

6 Sep 2024

PONE-D-23-38672R2Test-retest reliability of behavioral and computational measures of advice taking under volatilityPLOS ONE

Dear Dr. Karvelis,

Thank you for submitting your manuscript to PLOS ONE. After careful consideration, we feel that it has merit but does not fully meet PLOS ONE’s publication criteria as it currently stands. Therefore, we invite you to submit a revised version of the manuscript that addresses the points raised during the review process.

**Thank you for your valuable submission.**

1) Please double-check grammar (e.g. punctuation and wording);

2) Please double-check refs (e.g. abbreviations, page numbers, and sentence case need to be corrected);

3) Consider tidying Tables;

4) Please consider working on the graphs to ensure clarity:

- The use of clear marks indicating phase changes, and the use more distinct shading or color transitions to mark the "Stable" and "Volatile" periods are important;

- The layout of parameters in the HGF model (e.g. m3, k2) appears cluttered and could be streamlined;

- Increase the font size of axes labels and also ICC/r values for improved readability, ensuring the information is easily accessible;

- Make the CIs more distinct by increasing differences of the shading to better highlight uncertainty;

- Include some parameters values, such as the slopes. to give the reader more context about the relationship being depicted;

- Add a more detailed explanation for the importance of the “recovered” parameters and how they relate to the predictive performance. Clarify how deviations from the 1:1 line should be interpreted (e.g. whether significant deviations indicate model error).

error);

5) Conduct a power analysis to justify the sample size and enhance its validity;

6) Clarify the rationale for fixing certain model parameters and explain how this affects model flexibility and results interpretation;

7) Expand the discussion of practice effects and how they may have influenced outcomes during the retest, especially regarding their influence on reliability;

8) Provide a more detailed explanation of how parameter recoverability was assessed, including any limitations;

9) Consider incorporating cross-validation to confirm the generalisability;

10) Please elaborate on the potential influence of self-report bias on the conclusions and consider using objective measures where possible;

11) Consider allowing flexibility in the ϑ and ϕ parameters to better capture individual differences;

12) The current modeling framework, while robust, has some notable concerns. The fixed parameter values (e.g. meta-volatility and drift rate) lack clear justification, potentially limiting the flexibility and responsiveness to outcomes. Additionally, the rationale for the chosen priors for free parameters is unclear, raising concerns about their influence on outcomes and whether they sufficiently capture uncertainty;

- In light of these concerns, how do the fixed parameters constrain the ability to capture individual performance or outcomes differences? What specific influence might this have on prediction accuracy? Employing more flexible priors or diverse model selection methods are essential for reining adaptability and predictive accuracy;

- Could alternative modeling approaches offer a more accurate representation of learning trajectories?;

- Additionally, would longer intervals between sessions or more sophisticated model validation techniques strengthen the robustness of the findings?

13) Isn't it more intuitive to use Eq. 9 with a positive sign, considering volatility and the outcomes instead of ignoring uncertainty?;

- A more flexible approach would model learning rates dynamically, adapting to real-world strategies. Without these, the current model might miss aspects of real-world;

- Consider options beyond Gaussian distributions, as nonlinear uncertainty is a possibility;

- Why simplify the formula instead of using a robust method like GARCH for volatility?;

- The quasi-Newton optimisation algorithm was used to find parameter values with the highest posterior probability.  Consider validating whether it reaches the global maximum or risks local maxima;

- The authors could broader prior ranges to assess the robustness to different task dynamics;

- Examine correlation matrices to check if parameters are difficult to identify or highly correlated;

- Wouldn't CCC be more appropriate for checking linear relationships?;

- Why use IQR to detect outliers instead of z-scores? While filtering is useful, please clarify your choice;

- Include important reliability measures like B-A's coefficients and CV;

- Clarify that Pearson’s correlation is not a substitute for ICC in reliability evaluations;

- Explain why Bayesian Factors offer a better measure of evidence, particularly their ability to quantify uncertainty;

- Mention the key assumptions of paired t-tests and consider alternatives if assumptions are violated;

- Clarify what you mean by “extreme outliers” and explain how you differentiated between regular and extreme outliers, if applicable;

- Provide more detail on the “winning model” and how protected exceedance probability was calculated, explaining why it was used for model selection;

- Consider including potential explanations such as within-subject variability or measurement noise;

- Expand on how collinearity could affect parameter recoverability, and if ruled out, discuss other factors;

- Elaborate on the interpretation of Bayesian factors, explaining how values like 5.6 indicate "strong evidence" for the null hypothesis; 

- Clarify how outliers were handled and whether their removal significantly influenced the conclusions;

- Provide a thorough explanation of face validity and its role in assessing whether the task and model measure the intended constructs;

Wishing you success with the study.

We look forward to receiving your revised manuscript.

Kind regards,

Thiago P. Fernandes, PhD

Academic Editor

PLOS ONE
---

## [Author Response · Author response to Decision Letter 2]

28 Sep 2024

Thank you for providing clarification on the original comments. We carefully considered your suggestions and provided detailed responses.

---

## [Editor Report · Decision Letter 3]

4 Oct 2024

Test-retest reliability of behavioral and computational measures of advice taking under volatility

PONE-D-23-38672R3

Dear Dr. Karvelis,

We’re pleased to inform you that your manuscript has been judged scientifically suitable for publication and will be formally accepted for publication once it meets all outstanding technical requirements.

Kind regards,

Thiago P. Fernandes, PhD

Academic Editor

PLOS ONE
---

## [Editor Report · Acceptance letter]

6 Nov 2024

PONE-D-23-38672R3 

PLOS ONE

Dear Dr. Karvelis, 

I'm pleased to inform you that your manuscript has been deemed suitable for publication in PLOS ONE. Congratulations! Your manuscript is now being handed over to our production team.

Kind regards, 

on behalf of

Dr. Thiago P. Fernandes 

Academic Editor

PLOS ONE